# Predictive Model of Anxiety and Depression Perception in Multiple Sclerosis Patients: Possible Implications for Clinical Treatment

**DOI:** 10.3390/bioengineering11010100

**Published:** 2024-01-22

**Authors:** María Cuerda-Ballester, Antonio Bustos, David Sancho-Cantus, David Martínez-Rubio, Jesús Privado, Jorge Alarcón-Jiménez, Carlos Villarón-Casales, Nieves de Bernardo, Esther Navarro Illana, José Enrique de la Rubia Ortí

**Affiliations:** 1Doctoral Degree School, Catholic University San Vicente Mártir, 46001 Valencia, Spain; maria.cuerda@ucv.es; 2Physical Therapy Clinic, Antonio Bustos, 46007 Valencia, Spain; fisioterapiabustos@gmail.com; 3Department of Nursing, Catholic University San Vicente Mártir, 46001 Valencia, Spain; nieves.debernardo@ucv.es (N.d.B.); joseenrique.delarubi@ucv.es (J.E.d.l.R.O.); 4Department of Nursing and Physiotherapy, University of Lleida, 25006 Lleida, Spain; 5Department of Psychology, European University of Valencia, 46010 Valencia, Spain; 6Department of Methodology of Behavioral Sciences, Universidad Complutense de Madrid, Campus de Somosaguas, Pozuelo de Alarcón, 28223 Madrid, Spain; jesus.privado@pdi.ucm.es; 7Department of Physiotherapy, Universidad Católica de Valencia, 46900 Valencia, Spain; jorge.alarcon@ucv.es; 8Biomechanics & Physiotherapy in Sports (BIOCAPS), Faculty of Health Sciences, European University of Valencia, 46001 Valencia, Spain; carlosalberto.villaron@universidadeuropea.es

**Keywords:** multiple sclerosis, anxiety, depression, psychological well-being, functional activity, prefrontal activity

## Abstract

Multiple Sclerosis (MS) is a neurodegenerative disease characterized by motor and non-motor symptoms, including emotional distress, anxiety, and depression. These emotional symptoms currently have a pharmacological treatment with limited effectiveness; therefore, it is necessary to delve into their relationship with other psychological, functional, or prefrontal alterations. Additionally, exploring non-pharmacological therapeutic alternatives that have shown benefits in addressing emotional distress in MS patients is essential. Aim: To establish a predictive model for the presence of anxiety and depression in MS patients, based on variables such as psychological well-being, functional activity, and prefrontal symptoms. Additionally, this study aimed to propose non-pharmacological therapeutic alternatives based on this model. Materials and Methods: A descriptive, observational, and cross-sectional study was conducted with a sample of 64 diagnosed MS patients who underwent functional and cognitive assessments using the following questionnaires and scales: Functional Activities Questionnaire (FAQ), Acceptance and Action Questionnaire (AAQ-II), Experiences Questionnaire (EQ), Self-Compassion Scale Short Form (SCS-SF), Beck Depression Inventory II (BDI-II), State-Trait Anxiety Inventory (STAI), and Prefrontal Symptoms Inventory (PSI). Results: The model showed an excellent fit to the data and indicated that psychological well-being was the most significant predictor of the criteria (β = −0.83), followed by functional activity (β = −0.18) and prefrontal symptoms (β = 0.15). The latter two are negatively related to psychological well-being (β = −0.16 and β = −0.75, respectively). Conclusions: Low psychological well-being is the variable that most significantly predicts the presence of anxiety and depression in MS patients, followed by functional activity and prefrontal alterations. Interventions based on mindfulness and acceptance are recommended, along with nutritional interventions such as antioxidant-enriched ketogenic diets and moderate group physical exercise.

## 1. Introduction

Multiple Sclerosis (MS) is the most prevalent chronic disabling neurological disease in individuals over 18 years of age, affecting approximately 2.3 million people worldwide [1]. Spain has a medium-to-high prevalence, with 80–180 cases per 100,000 inhabitants [2]. The disease is characterized by a complex pathogenesis, including demyelination and involvement of axonal activity related to functionality [3]. The assessment of cognitive functions, with a prevalence of 50–60% [4], has been incorporated, with key predictors being the emotional distress experienced by patients, based on the presence of anxiety and depression [5]. Specifically, the most used pharmacological treatment for anxiety and depression is selective serotonin reuptake inhibitors (SSRIs), which show low efficacy and numerous side effects [6]. Additionally, the effectiveness of treatments for anxiety and depression is questionable [7,8]. For this reason, it seems interesting to establish which variables most predict the presence of depression and anxiety, to identify the most promising non-pharmacological therapeutic alternatives to alleviate these emotional disorders more effectively.

### 1.1. Emotional Disturbances, Anxiety and Depression

Mood disorders such as depression and anxiety are frequently observed in patients with MS, presenting a higher prevalence than that in the general population, and are two of the main comorbidities of the disease [9].

As the disease progresses, depression emerges as the predominant emotional disorder. Patients experiencing depression often exhibit lower psychosocial quality of life, reduced work capacity, or decreased therapeutic adherence [10]. 

Moreover, depression can contribute to the development of neurological symptoms inherent to MS and may also promote the onset of fatigue, memory disturbances, lack of concentration, insomnia, irritability, or loss of appetite. The most common alterations in anxiety include generalized anxiety, panic disorders, and obsessive compulsive disorder, all of which directly affect these patients’ quality of life [11]. 

### 1.2. Variables Related to Anxiety and Depression in MS

#### 1.2.1. Prefrontal Brian Activity

The onset of psychological changes can be observed even before the diagnosis of the disease, and patients may experience preliminary symptoms that progressively worsen [12]. Manifestations related to anxiety often occur in the initial stages, coinciding with the onset of symptoms and diagnosis [11]. Causal factors include inadequate social support and cognitive impairment. Anxiety can exacerbate MS symptoms such as fatigue or cognitive impairment, and it can have a negative impact on work, family, and social life [12]. 

At the prefrontal level, between 40 and 65% of patients with MS present cognitive and behavioral impairment, with the most affected cognitive functions being memory, attention, information processing speed, and various executive functions related to problem-solving [13]. In this regard, there is a relationship between elevated levels of anxiety and/or depression with a reduction in cognitive performance and attentional deficits [11,12], as well as deficits in metacognitive abilities [14]. 

#### 1.2.2. Functional Activity

In patients with MS, elevated levels of anxiety have been associated with a 2.9 times higher relapse rate [13], although health-related disabilities in mental health for MS patients are still not entirely clear. However, the link between loss of functional activity and emotional disturbances is evident. An increase in disability in individuals with MS is often associated with poorer psychosocial functioning, and there is evidence linking functional deterioration to psychiatric disorders such as anxiety and depression [12]. 

One of the primary limitations faced by patients with MS is gait disturbance. Regarding gait-related variables, the risk of falls in MS patients has been linked to motor deficits [15]. In this context, individuals with MS who are more physically active tend to have lower levels of depression, suggesting that poorer emotional health could negatively influence physical capacity [16]. These findings may explain the observed relationship between depression and loss of functional activity in patients with relapsing–remitting MS (RRMS) [17], with comorbid depression identified as a predictor of MS progression [18].

#### 1.2.3. Psychological Variables

Disease adaptation strategies, such as psychological flexibility, decentering, and self-compassion, play a crucial role in coping with and accepting the disease. These strategies are associated with reduced distress and lower intensity of negative emotions, making them essential factors in the overall coping process [19,20]. 

Psychological flexibility, or acceptance, is the ability to fully engage with the present moment as a conscious human being and to adapt or persist in behavior when aligned with valued goals [21]. This concept stands in direct contrast to experiential avoidance, which involves a conscious effort to evade contact with aversive private experiences and actively work to modify them or the conditions that give rise to them. This avoidance tendency, in turn, increases the risk of developing psychological issues such as depression or anxiety [21,22]. Additionally, acceptance has been linked to enhanced adaptation over time in individuals with MS, which is particularly crucial in the context of an unpredictable and progressive disease for which there is no cure [23].

Decentering is often defined as how an individual observes one’s thoughts and feelings as temporary, objective events in the mind, instead of personally identifying with them [24,25].

This distancing from mental content allows individuals to consider alternative perspectives, facilitating a conscious approach to challenges rather than simply reacting to or avoiding them [26]. This can assist people in becoming aware of recurring thoughts about the past or worrisome thoughts about the uncertain future related to the disease without necessarily becoming entangled in or pursuing them (which is particularly important in chronic conditions such as MS) [19]. 

Finally, compassion is a cognitive, affective, and behavioral process encompassing the recognition of suffering, acknowledgment of common humanity, empathy, tolerance for uncomfortable feelings, and motivation to alleviate the suffering of oneself and/or others [27]. As outlined by Neff (2003) [28], self-compassion comprises three key components: (1) practicing self-kindness in the face of suffering, (2) perceiving one’s experience as part of a broader human context, and (3) maintaining awareness of thoughts and feelings without excessive identification with them. Self-compassion is correlated with reduced levels of depression and anxiety, contributing to greater overall well-being [29,30]. 

### 1.3. Aim

Owing to the pharmacological limitations in the treatment of anxiety and depression, prevalent in a high percentage of patients with MS, the aim of the present study was to establish a predictive model of emotional distress based on the presence of anxiety and depression in patients with MS, utilizing variables such as psychological well-being, functional activity, and prefrontal symptoms. This model could also help propose non-pharmacological alternatives for the clinical improvement of MS symptoms.

## 2. Materials and Methods

### 2.1. Sample

The participants were patients with MS who belonged to different MS associations in the Valencia Region (Spain). Contact was established with the directors of these associations to disseminate the study and obtain the study sample, and the objectives of the study were explained. Subsequently, the directors contacted their members and provided them with patient information sheets. Those interested who signed the informed consent form were then subjected to the selection criteria. Inclusion criteria were as follows: patients over 18 years old diagnosed with RRMS, secondary progressive MS (SPMS), or primary progressive MS (PPMS) for at least one year, treated with glatiramer acetate and interferon-beta, and without a relapse in the last 6 months. Pregnant or lactating women, patients with dementia, those treated with antidepressants, and those with hormonal diseases affecting the hypothalamic–pituitary–adrenal (HPA) axis were excluded. A total of 64 patients diagnosed with MS using the McDonald criteria participated [31]. Six patients dropped out, resulting in a final sample of sixty-four participants with MS, with an average age of 46.8 years (SD = 11.93 years), of whom 28.1% were male. Among the patients, 68.8% had RRMS, 25.0% had SPMS, and 6.3% had PPMS. 

### 2.2. Instruments and Measurements

The Functional Activities Questionnaire (FAQ) [32,33,34] assesses the functional status of the patient through 11 items that evaluate basic and instrumental activities of daily living. It utilizes an ordinal scale of 4 points (0: independent; 1: performs independently but with difficulty; 2: requires assistance; 3: dependent). The score ranges from 0 to 30 points, and impairment is considered from a score of 9 onwards. This instrument has demonstrated high internal consistency in its various adaptations (α = 0.01–0.96).

The Acceptance and Action Questionnaire (AAQ-II) [35,36] comprises 7 items designed to measure experiential avoidance, which is conceptualized as the opposite of psychological flexibility, meaning psychological inflexibility. Psychological inflexibility refers to a conscious intention to avoid contact with private aversive experiences [21,22]. Each item is rated on a 7-point scale, with higher scores indicating a greater inclination toward experiential avoidance. The instrument demonstrated high internal consistency, with a Cronbach’s α = 0.89.

The Experiences Questionnaire (EQ) [37] is an 11-item questionnaire used to assess decentering, understood as the capacity to observe one’s thoughts and feelings as temporary and objective events of the mind. A high mean score indicates an elevated level of decentering. Items are rated on a five-point Likert scale (ranging from 1, never or very rarely true, to 5, very often or always true) with higher scores reflecting greater decentering. The Spanish version of the EQ showed high internal consistency (Cronbach’s α = 0.89). 

The Self-Compassion Scale Short Form (SCS-SF) [38,39] is a 12-item questionnaire used to evaluate overall self-compassion (total score) and components of self-compassion across three conceptually distinct yet theoretically interrelated facets: common humanity (SCS-CH), mindfulness (SCS-M), and self-kindness (SCS-SK). Despite the construct’s definition using these three facets, factor analysis suggested six subscales, capturing both the positive and negative aspects of each facet [28]. These items gauge how respondents perceive their actions during challenging times and are rated on a Likert-type scale from 1 (almost never) to 5 (almost always). The SCS demonstrates adequate reliability and validity, even across diverse cultures [40]. The SCS–SF exhibits satisfactory internal consistency (Cronbach’s α ≥ 0.86).

The Beck Depression Inventory II (BDI-II) [41,42] assesses depressive symptoms, primarily of a cognitive nature, while also considering physiological, emotional, or motivational manifestations. This version consists of 21 items and includes symptoms such as agitation, feelings of worthlessness, difficulty concentrating, and loss of energy. Each questionnaire item captures a depressive symptom, and for each item, four alternative statements are provided, ordered from the least to the most severe. Each item is rated on a scale of 0–3 points. In its Spanish version, the BDI-II demonstrated high internal consistency during validation, with an α value of 0.83.

The State-Trait Anxiety Inventory (STAI) [43,44] provides a measurement of both state anxiety and trait anxiety. The inventory consists of 40 questions, with 20 focusing on trait anxiety and 20 on state anxiety. All the questions are rated on a 4-point frequency scale. In its original version, the STAI demonstrated high internal consistency with a value of α = 0.88.

Inventory of Prefrontal Symptoms (IPS) [45]. This self-reported questionnaire explores cognitive, emotional, and behavioral disturbances in daily activities and is applicable to both the general population and various clinical populations. It comprises 20 items, scored on a Likert scale ranging from 0 (never or almost never) to 3 (always or almost always). The IPS showed good reliability, with α = 0.7–0.89.

### 2.3. Design and Procedure

This was a descriptive, observational, and cross-sectional study. The questionnaires were administered by nursing staff and neuropsychologists who are specialists in neurodegenerative diseases and members of the research team. It took the participants 20 min to complete the questionnaires and patients received instructions on how to complete them beforehand.

### 2.4. Statistical Analysis

First, the normal distribution of the data was calculated using SPSS version 21.

Second, confirmatory factor analyses were conducted using the AMOS V. 23 program [46] to examine whether behavioral, well-being, and physiological and emotional distress measurements were theoretically clustered into a latent factor. To assess the fit of the data to these tested models, two types of goodness-of-fit indices were employed: (1) Absolute indices, to determine if the theoretical model fits the empirical data, including the index χ^2^/df [47], whose values below 3 indicate a good fit; the goodness-of-fit index (GFI) [48], with values >0.95 considered a good fit; and the Standardized Root Mean Square (SRMR) [49] and Root Mean Squared Error (RMSEA) [50], with values <0.08 indicating a good fit [51]. Additionally, the presence of < 5.00% standardized residuals exceeding 2.58 in absolute value was considered [51]; (2) Incremental indices were utilized to compare the obtained model with the null model. These include the Normed Fit Index (NFI) [47], Comparative Fit Index (CFI) [52], and Tuker–Lewis Index (TLI) [53], with values >0.95 indicating a good fit. It is recommended to have at least 10 participants per indicator for factor analyses [54], although some suggest only 5 per indicator when the distribution is normal [51]. In this case, we met the latter criterion with 64 participants for three to six indicators across the different tested models (64/3 = 21.33; 64/6 = 10.67). Furthermore, MacCallum et al. [55] conducted numerous simulations with varying communalities, the ratio between indicators and extracted factors, and sample size. They found, for *n* = 60, a convergence of 74.6% between empirical and population factors for ratios of 10/3 = 3.33 (indicators/factors) for communalities between 0.20 and 0.40. In our case, the ratios for the tested models ranged between 3 and 4, and most of the extracted communalities were suitable for the models (see Table 1) with *n* = 64, thus satisfying these. 

Finally, a predictive model was tested using various measurements from patients with MS, including functional and psychological factors, to predict emotional distress (anxiety and depression). The factors estimated from the confirmatory factor analyses (functional, well-being, and emotional distress) were utilized, with the addition of prefrontal symptom data. This addition aimed to reduce the number of measurements in the model and meet the criterion of at least 10 subjects per measurement. Specifically, six variables were employed for *n* = 64, resulting in a ratio of 64/6 = 10.67.

In Figure 1, the predictive model of emotional distress in patients with MS is summarized.

### 2.5. Ethical Considerations

This study was conducted according to the principles of the Declaration of Helsinki [56] and was approved by from the Human Research Committee of the University of Valencia (procedure number H1512345043343). The patients included in the study provided informed consent after being briefed on the procedures and the nature of the study.

## 3. Results

### 3.1. Descriptive Analysis

Table 1 presents the descriptive statistics of the data collected from the patients with MS. When skewness values do not exceed 2 and kurtosis values do not exceed 7 in absolute terms, a normal distribution of the variable is assumed. This is a prerequisite for using the maximum likelihood procedure to estimate the confirmatory factor and predictive models [57]. All measures exhibited a normal distribution.

### 3.2. Confirmatory Factor Analysis 

Two Confirmatory Factor Analyses were conducted to extract four latent factors: well-being and emotional distress. Figure 1 displays the estimated models and Table 2 presents the goodness-of-fit indices.

Both models exhibit multivariate normality, and they were estimated using maximum likelihood. Model a (well-being) shows multivariate normality through the Bollen–Stine bootstrap (*p* = 0.119) and fits the data well, with only two goodness-of-fit indices having non-recommended values (TLI = 0.866 and RMSEA = 0.167). Additionally, the factor loadings are above the recommended threshold of |≥0.40|. On the other hand, model b (emotional distress) does not show multivariate normality according to the Bollen–Stine bootstrap (*p* = 0.015). However, kurtosis is 1.15 (less than 7 in absolute terms), allowing the assumption of multivariate normality. Moreover, its factor loadings are above the minimum recommended value. The goodness-of-fit indices indicate a moderate fit to the data. As seen in Figure 2, two latent factors (psychological well-being and emotional distress) were identified, each consisting of at least three indicators that provide structural validity to these factors. These factors will be utilized in subsequent analyses.

Finally, Table 1 shows the communalities (*h*^2^) for the various indicators of the confirmatory models. Most of them fall within the minimum recommended range (between 0.20 and 0.40) by MacCallum et al. [55], with many even significantly exceeding this range. This result ensures that the findings remain fairly stable despite the small sample size (*n* = 64).

### 3.3. Predictive Model of Emotional Distress

Due to a small sample size of only 64 participants, it was not possible to estimate a predictive model that considered the three factors obtained for well-being and emotional distress separately. Instead, the latent factors of these variables were estimated and included as a single variable in the model. A model was estimated using these two predictors and prefrontal symptoms to predict emotional distress. Figure 3 displays the estimated model, and Table 2 presents the goodness-of-fit indices.

The predictive model exhibited estimated multivariate normality through the Bollen–Stine bootstrap (*p* = 0.289) [58], and the three predictors collectively account for 93% of emotional distress (*R*^2^ = 0.93). The model fits the data well. Psychological well-being is the strongest predictor of the criterion (β = −0.83), followed by functional activity (β = −0.18) and prefrontal symptoms (β = 0.15) with similar values. Therefore, higher psychological well-being corresponds to lower emotional distress in these individuals. Additionally, the model reveals a clear negative association between the presence of prefrontal symptoms and psychological well-being (β = −0.75), as well as a negative association between functional activity and well-being (β = −0.16), to a lesser extent. Therefore, emotional distress primarily decreases when psychological well-being is elevated, and functional activity and prefrontal symptoms play a lesser role in influencing the outcomes. While functional activity has a slightly reducing effect on emotional distress, prefrontal symptoms tend to elevate it.

## 4. Discussion

This study aimed to determine which of the analyzed variables, associated with the presence of emotional distress, had greater predictive value in the perception of anxiety and depression commonly found in most patients with MS. Therefore, functional activity, psychological well-being, and prefrontal symptoms were examined in a population of 64 patients.

### 4.1. Prefrontal Symptoms

Like previous studies [59,60], the model suggests a higher presence of emotional distress (manifested by anxiety and depression) with increased prefrontal symptomatology. In this sense, Wallis et al. noted that anxiety, depression, or mental fatigue represented a substantial part of the total variance of cognitive problems (manifested by prefrontal symptoms) in patients with MS. 

Additionally, Beier et al. [61] demonstrated that factors such as anxiety or fatigue acted as predictors of prefrontal symptomatology, also in a sample of individuals diagnosed with MS. Leavitt et al. supported this line of evidence, confirming the relationship between cognition and anxious–depressive manifestations in MS [60]. 

However, despite the alignment between our results and those previously published, it is worth noting that prefrontal symptoms predict emotional distress, but to a limited extent (β = 0.15). In contrast, the relationship appears to be more significant in previous studies. This difference could be attributed to variations in the diagnosis of the study population. For instance, in the population studied by Beier et al. [61], the diagnosis was self-reported and also included patients with Progressive Relapsing MS (PRMS), a subgroup that was not included in our study. On the other hand, Leavitt et al. [60] studied a sample of newly diagnosed patients, while the patients in our study had been diagnosed with MS for at least 12 months.

### 4.2. Functional Activity

Similar results were obtained for functional activity. According to our findings, functional activity negatively predicts the criterion, meaning that higher functional activity is associated with a lower presence of depression and anxiety, with a slightly higher weight than that shown by prefrontal alterations (β = −0.18). This result confirms recent observations in the study by Lo Buono V et al. [62], where functional activity was identified as a significant predictor for anxiety and depression subscales, among others, as observed using the Symptoms Checklist-90-Revised (SCL-90-R) questionnaire. However, these findings are not aligned with those obtained by Alswat et al. [63], where no association was observed between functional activity and the presence of depression and anxiety in patients with MS from Saudi Arabia. As the authors explain, this discrepancy could be attributed to the fact that their study employs the Expanded Disability Status Scale (EDSS) to measure capacity. The EDSS primarily focuses on physical disability and motor system dysfunction rather than directly assessing mood, cognitive manifestations, or psychiatric conditions [64]. 

Therefore, it seems evident that there is an association between functional activity and these emotional variables, such as depression, as previously demonstrated in 2003 by Peres et al. [65]. This association further explains the deterioration of physical health [66]. 

### 4.3. Psychological Well-Being

Based on this study, it is finally worth highlighting that the primary positive predictor of emotional distress in patients with MS is psychological well-being (β = −0.87). Consistent with previous findings, decentering and experiential avoidance (EA) emerge as core elements in patients with chronic pain [67]. 

Regarding EA, its impact on mental health has been studied by various authors, for instance, in disorders such as generalized anxiety or panic disorder [68,69]. Concerning depression, half of the variation in depressive symptomatology could be attributed to the lack of acceptance of painful experiences. This relationship has also been observed in diabetes, physical pain, and spinal cord injury, among others [70]. In the case of physical pain, EA plays a mediating role in the relationship between self-compassion and depressive symptoms, even after controlling for the effect of different coping strategies [71]. 

In line with our results, other authors have noted a negative relationship between EA and emotional distress variables (depression and anxiety). Participants find it challenging to maintain difficult thoughts and emotions, which is why they opt for this type of coping that opposes acceptance [72]. Looking at the negative facets of the Self-Compassion Scale (SCS), it is observed that higher levels of EA are associated with greater self-identification, self-criticism, and isolation [19]. The results of this study emphasize the importance of preventing emotional distress through interventions aimed at promoting psychological flexibility and minimizing EA [73,74]. 

Regarding decentering, higher scores are related to greater well-being or less emotional distress and a reduction in depressive symptoms. Previous studies indicate that developing a metacognitive insight promotes the observation of thoughts and emotions as passing mental events rather than definitive products of oneself [75], demonstrating the durability and positive effect of intervention in the development of metacognitive abilities for depression [76] and anxiety [77]. Additionally, decentering is contrary to rumination and avoidance strategies, which are closely related to the development of depressive and anxious symptoms [78,79]. The capacity for decentering has been highlighted as an effective coping strategy in patients with chronic illnesses [79], improving quality of life.

Regarding self-compassion, in line with Bogosian et al. [19], higher scores are associated with lower depression and anxiety and greater well-being, probably due to a kinder view of oneself. This allows them to treat themselves with kindness and see their feelings without judgment [28]. In a sample of patients undergoing hemodialysis, a relationship between self-compassion and the presence of positive emotions, as well as greater well-being and less EA, was observed [80]. These findings are consistent with other samples of cancer patients, where a reduction in depression and stress was also observed. Pinto-Gouveia also agrees with the results obtained in this study, stating that higher self-compassion significantly predicts lower levels of depressive symptoms and stress, as well as better quality of life [81,82]. Ozonder [83] further asserts that, in cancer patients, the presence of self-compassion acts as a protective factor against depressive disorders. Therefore, individuals diagnosed with cancer had lower scores on some SCS subscales, such as self-compassion, common humanity, and mindfulness, in contrast to dimensions such as self-judgment, isolation, or over-identification. All these elements could, in turn, influence the presence of greater EA and, therefore, poorer acceptance of the disease. Regarding anxiety, those individuals who showed self-compassion were more likely to develop protective factors for anxiety. Both our results and those of the mentioned authors are consistent with studies conducted on patients with MS [84]. 

The presence of a self-compassionate approach could limit the occurrence of maladaptive cognitive processes related to depressive symptomatology, promoting adaptive strategies such as acceptance [28] or resilience [74]. These responses could help patients overcome unpleasant events and develop positive emotions. 

### 4.4. Clinical Implications

Clinically, given the significant impact of psychological well-being on the perception of anxiety and depression in this condition, it can be asserted that the model provides a theoretical basis for understanding some of the psychological mechanisms underlying emotional distress in patients with MS. This allows us to identify key factors in the development of depressive and anxious symptomatology related to MS, such as EA, isolation, self-criticism, or over-identification. Additionally, it facilitates the development of specific individual and group psychological interventions aimed at improving psychological flexibility, decentering, self-kindness, shared humanity, or mindfulness skills. Specifically, in patients with MS, mindfulness-based interventions have shown a moderate effect in improving fatigue and high effectiveness in reducing stress after intervention, although these findings are attenuated for fatigue during follow-up [85]. Group intervention programs that include these elements could be valid alternatives in these populations, including both Mindfulness-Based Interventions (MBIs) [19] and Mindfulness-Based Stress Reduction (MBSR) [24,86]. Complementary to this, understanding how psychological variables impact emotional distress can contribute to developing strategies that improve adaptation and adherence in patients with MS and similar chronic conditions such as rheumatoid arthritis, inflammatory bowel disease, pain, or diabetes, as psychological aspects also play a crucial role in disease management [23]. This involves accepting daily challenges associated with MS, managing their own emotions (recognizing thoughts and feelings), and focusing on the present moment to reduce the mentioned symptoms.

The proposed psychological interventions can be complemented by trying to improve the other two variables (functional activity and prefrontal symptoms). As mentioned earlier, both have a lower predictive value than psychological well-being, but in any case, they have a clear relationship with the presence of depression and anxiety according to the model. Additionally, we observe a clear association between the presence of prefrontal symptoms (β = −0.75) and functional activity (β = −0.16) from our results (Figure 2). Therefore, it is essential to highlight the relationship between nutritional habits and the presence of depression [87], which could explain the improvements in the perception of depression observed in our laboratory after administering a diet that promotes ketogenesis supplemented with polyphenols, which managed to improve cortisol activity linked to depression [88]. Moreover, this same treatment managed to reduce anxiety significantly, while achieving a significant decrease in functional activity [89], identifying nutritional treatments as an alternative for emotional improvements through the reduction in functional activity. Additionally, it is interesting to note that this same type of intervention based on polyphenol administration also shows cognitive and behavioral improvements related to prefrontal activity [90,91].

In addition to diet, exercise has a very positive impact on preventing functional loss, as well as improving emotional aspects associated with a higher quality of life [92,93]. Thus, physical exercise is linked to a lower occurrence of relapses and neurological improvements [94], especially highlighting the practice of moderate group exercise [95,96,97].

Moreover, considering the relationship between prefrontal symptoms and cognitive functions, it would be interesting for future studies to assess the different executive functions using specific tools such as the Seven Minute Screen [98] composed of the following tools: Benton’s temporal orientation test, the Clock Drawing Test (assessing visuospatial and visuoconstructive abilities), the Categorical Verbal Fluency Test, assessing semantic memory, and the Selective Recall Test to assess episodic memory, or the Trail Making Test (TMT) [99], a neuropsychological test assessing processing speed, attention, working memory, and cognitive flexibility. It consists of two parts: Part A, in which the examinee must connect 25 numbered circles in ascending order, and Part B, where the examinee must alternate between circles with numbers and letters in alphabetical and numerical order. The time taken to complete each part is the performance indicator.

The main limitation of this study is the small sample size (*n* = 64), which affects the validity of the conclusions. Therefore, it would be interesting to replicate these results in larger MS samples. Despite this small sample size, the sizes of the factorial weights and communalities of the different estimated models present adequate values, which ensures the integrity of our results [55]. Another limitation is the use of a cross-sectional design, so future studies with longitudinal designs should verify the consistency of our results. Additionally, specific executive functions could have been assessed, which would provide greater depth to the findings regarding prefrontal symptomatology.

## 5. Conclusions

This study presents innovative work using a predictive methodology based on structural equation models, aiming to reliably predict emotional distress in small samples from various predictors (functional and psychological) in individuals with MS. 

Based on these results, it may be concluded that low psychological well-being is the variable that most predicts the presence of anxiety and depression in patients with MS, followed by functional activity, with prefrontal alterations carrying less weight. Based on these conclusions, and after analyzing non-pharmacological therapeutic alternatives from other published studies, it appears that the model supports Mindfulness and Acceptance-Based Interventions in patients with MS as effective interventions in reducing depressive and anxious symptomatology. These therapies could be complemented with nutritional interventions such as antioxidant-enriched ketogenic diets and moderate group exercise.

## Figures and Tables

**Figure 1 bioengineering-11-00100-f001:**
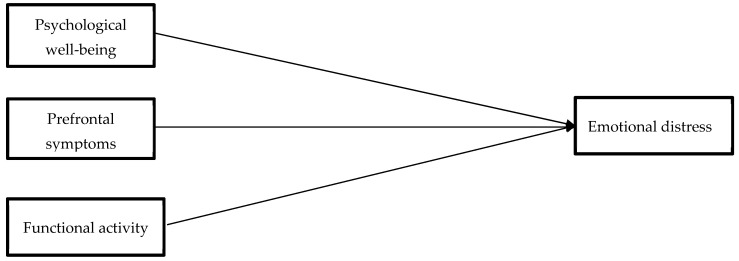
Predictive model of emotional distress in patients with Multiple Sclerosis proposed in the study.

**Figure 2 bioengineering-11-00100-f002:**
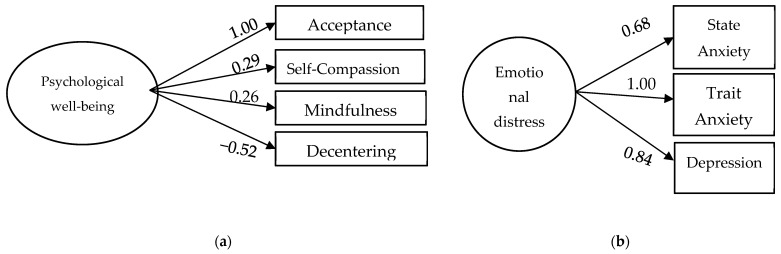
Confirmatory Factor Analyses for (**a**) psychological well-being, and (**b**) emotional distress (anxiety and depression).

**Figure 3 bioengineering-11-00100-f003:**
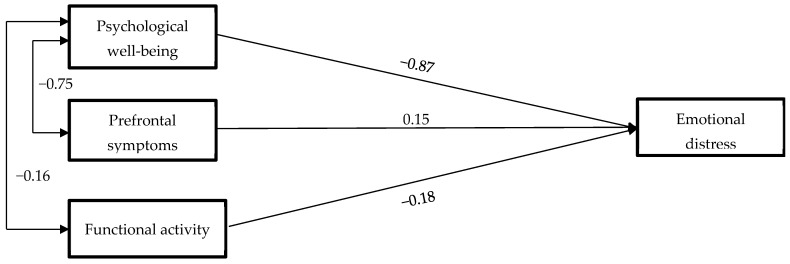
Predictive model of emotional distress based on psychological well-being, functional activity, and prefrontal symptoms. Note: Correlations lower than |± 0.15| were removed to simplify the model.

**Table 1 bioengineering-11-00100-t001:** Descriptive statistics, distribution, and communality (*h*^2^) of the collected data.

Measurements	M	SD	Asymmetry	Kurtosis	*h*^2^ Models
Functional Activity	5.77	8.67	1.92	3.10	
Acceptance Action Questionnaire	23.98	10.24	0.42	−0.62	1.00
Self-Compassion Scale	36.90	6.33	−0.63	1.83	0.08
Mindfulness Questionnaire	44.50	7.52	−0.39	0.99	0.07
Experiences Questionnaire	38.41	7.52	0.05	0.18	0.27
Inventory of Prefrontal Symptoms	54.48	27.91	0.28	−0.65	
State Anxiety	20.02	13.07	0.78	0.38	0.46
Trait Anxiety	25.64	12.69	0.33	−0.71	1.00
Depression	13.52	9.82	0.76	−0.21	0.71

**Table 2 bioengineering-11-00100-t002:** Goodness-of-fit indices for Confirmatory Factor Analyses and predictive models.

	χ^2^/df	GFI	NFI	CFI	TLI	RMSEA	SRMR	Residues ≥ |±2.58|
Confirmatory model a	2.761	0.979	0.968	0.978	0.866	0.167	0.090	0.00%
Confirmatory model b	9.314	0.916	0.926	0.933	0.798	0.363	0.060	0.00%
Predictive model	1.981	0.985	0.992	0.996	0.977	0.125	0.082	0.00%

## Data Availability

Data are contained within the article.

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
