# Peer review of "Predictive Model of Anxiety and Depression Perception in Multiple Sclerosis Patients: Possible Implications for Clinical Treatment"

_bioengineering, 2024, doi:10.3390/bioengineering11010100_

Round 1
Reviewer 1 Report
Comments and Suggestions for Authors
The authors presented the work titled as "Predictive Model of Anxiety and Depression Perception in Multiple Sclerosis Patients. Possible Implications for Clinical Treatment."
There are some comments which need to be addressed:
In title replace "Patients. P" with "Patients: P"
In abstract first line must correct (EM) to (MS)
please check if these abbreviations are correct or not: Acceptance and Action Scale (AAQ-II),
On page this paragraph needs better quality english "Of the two emotional disorders, depression..."
In method section, It would be quite helpful if authors could summarize the entire work into a simplified small mathematical model for better understanding and clarity. if not then please present an integrated workflow.
There are several places where the typos mistakes are because of merging of the alphabets or characters. Please rectify it.
Acceptance and Action Scale (AAQ-II)
Comments on the Quality of English LanguageThere is still need for english improvement, please go through it.
Author Response
Reviewer 1
The authors presented the work titled as "Predictive Model of Anxiety and Depression Perception in Multiple Sclerosis Patients. Possible Implications for Clinical Treatment."
There are some comments which need to be addressed:
1.-In title replace "Patients. P" with "Patients: P"
It has been replaced, as suggested. Thank you
2.-In abstract first line must correct (EM) to (MS)
It has been amended, apologies. Thank you
3.-Please check if these abbreviations are correct or not: Acceptance and Action Scale (AAQ-II),
The abbreviation is: AAQ-II is Acceptance and Action Questionnaire- II (AAQ-II). It has been changed in the Abstract and in Methods.
Original reference: Bond, F. W., Hayes, S. C., Baer, R. A., Carpenter, K. M., Guenole, N.,Orcutt, H. K., . . . Zettle, R. D. (2011). Preliminary psychometric properties of the Acceptance and Action Questionnaire – II: A revised measure of psychological inflexibility and experiential avoidance. Behavior Therapy, 42, 676-688.
Spanish validation: Ruiz, F. J., Langer, A. I., Luciano, C., Cangas, A. J., & Beltrán, I. (2013). Measuring experiential avoidance and psychological inflexibility: The Spanish translation of the Acceptance and Action Questionnaire – II. Psicothema, 25, 123-129.
On page this paragraph needs better quality english "Of the two emotional disorders, depression..."
As recommended, the sentence has been rephrased. Thank you
In method section, It would be quite helpful if authors could summarize the entire work into a simplified small mathematical model for better understanding and clarity. if not then please present an integrated workflow.
Thank you for your comment. As suggested by the reviewer, we have added Figure 1, which depicts the predictive model we are trying to validate in order to predict emotional distress in patients with multiple sclerosis, in order to clarify.
There are several places where the typos mistakes are because of merging of the alphabets or characters. Please rectify it.
As pointed out, this has all been reviewed and corrected. Apologies and thank you for the observation
Reviewer 2 Report
Comments and Suggestions for Authors
The paper presents a well-structured study exploring the predictors of emotional distress in patients with Multiple Sclerosis (MS). The use of structural equation modeling provides a robust analytical approach. The discussion on clinical implications and potential interventions based on the study's findings is thorough and valuable. Linking psychological interventions, nutritional approaches, and physical exercise to the predictors identified in the study provides practical insights for clinicians
However, there are some suggestions for improvement and considerations for acceptance.
While the study provides valuable insights, the sample size (n = 64) is relatively small. The authors should acknowledge this limitation and discuss its potential impact on the generalizability of the findings (even though sample size was computed before the study). Consideration should be given to replicating the study with a larger sample.
Considering the link between prefrontal symptoms and cognitive functions, it would be beneficial to include assessments of specific executive functions. This could enhance the understanding of the relationship between prefrontal symptoms and emotional distress.
The authors should clarify the statement that the model fits the data "very well." Providing specific fit indices and comparisons with alternative models would strengthen this claim through the results section.
Consider expanding the discussion section to delve deeper into potential confounding variables or alternative explanations for the observed relationships. This would enhance the paper's overall robustness.
Minor point:
Abstract: Multiple Sclerosis (MS) and not EM
Table 1: Functional activity (typo)
Comments on the Quality of English LanguageGood
Author Response
Reviewer 2
The paper presents a well-structured study exploring the predictors of emotional distress in patients with Multiple Sclerosis (MS). The use of structural equation modeling provides a robust analytical approach. The discussion on clinical implications and potential interventions based on the study's findings is thorough and valuable. Linking psychological interventions, nutritional approaches, and physical exercise to the predictors identified in the study provides practical insights for clinicians
However, there are some suggestions for improvement and considerations for acceptance.
While the study provides valuable insights, the sample size (n = 64) is relatively small. The authors should acknowledge this limitation and discuss its potential impact on the generalizability of the findings (even though sample size was computed before the study). Consideration should be given to replicating the study with a larger sample.
Thank you for your comment. In response to the reviewer, we have clarified and addressed that limitation, along with its impact on the conclusions.Principio del formulario
Considering the link between prefrontal symptoms and cognitive functions, it would be beneficial to include assessments of specific executive functions. This could enhance the understanding of the relationship between prefrontal symptoms and emotional distress.
Considering the reviewer's considerations, specific information regarding a future evaluation of cognitive function has been added
The authors should clarify the statement that the model fits the data "very well." Providing specific fit indices and comparisons with alternative models would strengthen this claim through the results section.
We appreciate the reviewer's concern, as the aspect mentioned is crucial for the quality of the article. We believe that this information is detailed in section 2.4 Statistical Analysis, 2nd paragraph. However, if the reviewer wishes for us to add anything more, please let us know, and we will make effort to enhance it
Consider expanding the discussion section to delve deeper into potential confounding variables or alternative explanations for the observed relationships. This would enhance the paper's overall robustness.
Following your instructions, we have tried to enhance the discussion, specifically focusing on the relationship between emotional distress, prefrontal symptoms, and functional activity. We realize that that improvement was necessary in that section, and we truly appreciate your feedback.
Minor point:
Abstract: Multiple Sclerosis (MS) and not EM
Mistake amended. Thank you for the observation
Table 1: Functional activity (typo)
Likewise, for Table 1, the amendment has been carried out.
Reviewer 3 Report
Comments and Suggestions for Authors
1. In the abstract, the abbreviation for multiple sclerosis is MS, not EM, and AAQ-II does not stand for Acceptance and Action Scale.
2. In the abstract, all abbreviations need to be revised.
3. In the Materials and Methods section, what disclosure criteria were used?
4. In Table 1, the data needs revision for statistical analysis.
5. Where is the raw data for measurements located?
Comments on the Quality of English Language
1. Grammar and punctuation should be reviewed and revised.
Author Response
Reviewer 3
- In the abstract, the abbreviation for multiple sclerosis is MS, not EM, and AAQ-II does not stand for Acceptance and Action Scale.
Following the reviewer’s comment, this has been addressed. Thank you for the observations.
- In the abstract, all abbreviations need to be revised.
Following the reviewer’s comment, this has also been addressed (MS, AAQ-II and PSI). Thank you for your help.
- In the Materials and Methods section, what disclosure criteria were used?
In response to the reviewer, this has been added.
- In Table 1, the data needs revision for statistical analysis.
Thank you for your comment. Table 1 presents a descriptive analysis to show descriptive statistics and the distribution of the measured variables. This is why there is no inferential analysis. However, if the reviewer requires any additional analysis, we await further indications
- Where is the raw data for measurements located?
Given the confidentiality specifications of patient data by the ethics committee approved (December 14, 2017), Human Research Committee of the University of Valencia, the data were not shared on any platform. This database is exclusively available to the head researcher of the study and the statistician
Reviewer 4 Report
Comments and Suggestions for Authors
Predictive Model of Anxiety and Depression Perception in Multiple Sclerosis Patients. Possible Implications for Clinical Treatment.
Maria et al has concluded that low psychological well-being is the variable that most predicts the presence of anxiety and depression in MS patients, followed by functional 40 activity and prefrontal alterations. Interventions based on Mindfulness and Acceptance are recommended, along with nutritional interventions such as antioxidant-enriched ketogenic diets and moderate group physical exercise
The article is very interesting and finds a constructive model for Multiple Sclerosis Patients. I recommend the publication of the article after minor revision.
Abstract:
· Try to more deeply write the background of the study.
Main body:
· Can you update the references? According to the study the references should be updated.
· Explain the aim of the study.
· Plagiarism of a research article should not more than 17%, try to minimize the similarity index.
· Try to improve the conclusion
Comments on the Quality of English LanguageEnglish is fine
Author Response
Reviewer 4
Predictive Model of Anxiety and Depression Perception in Multiple Sclerosis Patients. Possible Implications for Clinical Treatment.
Maria et al has concluded that low psychological well-being is the variable that most predicts the presence of anxiety and depression in MS patients, followed by functional 40 activity and prefrontal alterations. Interventions based on Mindfulness and Acceptance are recommended, along with nutritional interventions such as antioxidant-enriched ketogenic diets and moderate group physical exercise
The article is very interesting and finds a constructive model for Multiple Sclerosis Patients. I recommend the publication of the article after minor revision.
Abstract:
- Try to more deeply write the background of the study.
We have attempted to expand that section while considering the word limit. If you have any further suggestions or specific areas for improvement, please let us know. We thank you for the comment
Main body:
- Can you update the references? According to the study the references should be updated.
In response to the reviewer's feedback, we have updated references in the manuscript. We appreciate the reviewer's request, as we believe it has enhanced the quality of the article
- Explain the aim of the study.
In response to your request, we have attempted to clarify the objective. Additionally, in the methodology section, we have added Figure 1 as recommended by another reviewer.
- Plagiarism of a research article should not more than 17%, try to minimize the similarity index.
We have analyzed the entire article and some paragraphs have been rewritten to try and decrease this percentage, although the paper is entirely original
- Try to improve the conclusion
We truly thank you for your request. In response to the reviewer, we have tried to enhance that section of the article, specifically in the areas where a more in-depth analysis is indeed necessary, such as the relationship of emotional distress with Prefrontal Symptoms and Functional Activity
Round 2
Reviewer 3 Report
Comments and Suggestions for Authors
Results needed to be more detailed
Comments on the Quality of English LanguageGrammar requires some revision
Author Response
Reviewer 3
Results needed to be more detailed
Following the reviewer's instructions, additional information has been added in the Results section for better understanding
Grammar requires some revision
Thank you for the comment. We proceed to review the entire text once again so as not to miss any grammatical mistakes